# Does previous circumcision and wealth index influence women's attitude to discontinue the practice of female genital mutilation and cutting (FGM/C) in Ethiopia?

**Mohammed Ahmed** [1] *, **Abdu Seid**[2], **Seada Seid**[1], **Ali Yimer**[1]

**1** Department of Public Health, College of Health Science, Woldia University, Woldia, Ethiopia, **2** Department of Midwifery, College of Health Science, Woldia University, Woldia, Ethiopia

* mohaasrar12@gmail.com

**Data Availability Statement:** We used the USAID–DHS program 2016 Ethiopian demographic and health survey data set for this analysis. To request the same or different data for another purpose, a

## Abstract

### Introduction

understanding women's attitudes towards female genital mutilation is an important step towards eliminating this practice. We used the 2016 Ethiopia Demographic and Health Survey (EDHS) data set to examine the relationship between wealth index, and previous history of circumcision on women's opinions whether female genital mutilation (FGM) should be continued or stopped in Ethiopia.

### Methods

Data from 6984 women aged 15–49 years were extracted from the 2016 Ethiopia EDHS data set. Multivariable logistic regression analysis was performed to analyse the data.

### Result

In this study, women with a higher level of education and wealth index were more likely to support the cessation of FGM. However, circumcised women (AOR: 0.22; 95% CI: 0.15–0.32), women from the Afar region (AOR: 0.34; 95% CI: 0.22–0.50), Somali region (AOR: 0.42; 95% CI: 0.27–0.65), and Dire Dawa region (AOR: 0.51; 95% CI: 0.32–0.83) were less likely to support discontinuation of FGM.

### Conclusion

The present study revealed that wealth index, education level, history of circumcision, and regional variation are associated with women's attitude towards discontinuation of the practice of FGM in Ethiopia. Empowering women in terms of socioeconomic status and education can change attitudes and might help prevent female genital mutilation in the future. Furthermore, interventions targeting FGM practices should focus on regional variance in order to have a meaningful impact on reducing this harmful cultural practice in Ethiopia.

new research project request should be submitted to the DHS program here: https://dhsprogram.com/data/Access-Instructions.cfm. After receiving permission, the researcher can log in and select the specific data in the format they prefer.

**Funding:** The author(s) received no specific funding for this work.

**Competing interests:** The authors have declared that no competing interests exist.

# Background

Female genital mutilation/cutting (FGM) is defined as any procedures that involve partial or entire removal of the external female genitalia or other harm to the female genital organs for non-medical reasons [1]. The prevalence of FGM become decreased in Ethiopia over the last 16 years, with prevalence declining from 80% in the 2000 Ethiopia Demographic and Health Survey (EDHS) to 74% in 2005, and 65% in 2016 EDHS [2]. However, the prevalence varies by regions: 99% in Somali, 91% in Afar, 33% in Gambela, and 24% in Tigray [2].

FGM is a dangerous procedure that predates all faiths and is carried out on children for a variety of reasons in different cultural and global contexts [3], resulting in acute and long-term repercussion's such as bleeding, shock, urine retention, and infertility [4–7]. In addition, according to a study conducted in Ghana, FGM was linked to an 8.2% caesarean section rate compared to 6.7% in mothers who did not have FGM [8].

The first step toward ending FGM is to change people's perceptions about it, which can be tough and psychologically painful [9]. In Ethiopia, 21% percent of women think that the practice should be continued [2]. Studies showed that attitude towards cessation of FGM was attributed to different factors such as male sex [10, 11], urban residence [12], educational attainment [5, 12–14], and access to media [12, 13]. In contrast, women with positive cultural conceptions of FGM, being muslim religion follower [12, 13, 15], 15–24 years age, living in rural areas, and being married [13] were less likely to support cessation of FGM.

When it comes to the impact of previous circumcision and socioeconomic status on attitudes towards FGM, the evidence is mixed. For example, one study in Egypt showed that previous circumcision favors discontinuation [11], while another study conducted in Ethiopia found that previous circumcision leads to the continuation of FGM [13]. Similarly, a higher levels of household wealth increased women's support for discontinuation [16]. In some countries, wealth index was linked to support FGM, and in some countries the opposite is true [17].

Previous research in Ethiopia regarding the continuation of female genital mutilation [13, 18] did not include crucial factor such as wealth index, which is an important to alleviate the barriers for discontinuation of FGM. Therefore, the current study sought to fill this vacuum by examining the relationship between wealth index, and previous history of circumcision on women's attitude to end the practice of FGM in Ethiopia, utilizing data from the country's most recent 2016 national demographic and health survey.

# Methods

## Data source

The current research used the 2016 Ethiopia Demographic and Health Surveys (EDHS) data set. A stratified cluster sampling method with two stages was applied. A detailed description of the study design and survey methods may be read elsewhere [2]. In the 2016 EDHS, 15,683 women (15–49 years) were interviewed, with a 95% response rate. After excluding cases with missing values on the variables of interests, the analytic sample in this study consisted of 6984 reproductive age women (15–49 years).

## Study variables

The dependent variable was the attitude towards the discontinuation of female genital mutilation, which was measured by asking the questions "circumcision should be continued or stopped?". There were two options for the respondents: continue or stop. The main exposure variables were whether or not women had ever been circumcised (yes or no) and the wealth index (a composite index based on the household's ownership of several consumer items),

which had five categories (poorest, poorer, middle, richer, and richest) according to the data set [2].

Covariates that affect attitudes towards the discontinuation of female genital mutilation include respondent's age, religion, marital status, education status, type of residence, access to media, and regions.

## Statistical analysis

SPSS version 21 was used to analyse the data. Frequencies and weighted percentages of study variables were reported. Bivariate analysis using Rao–Scott chi-square test was performed to examine the relationship between attitude towards the discontinuation of female genital mutilation and each of the independent variables and to select potential candidates for the multivariable model. A variable, which had a p-value of < 0.25 in bivariate analysis were entered into a multivariable logistic regression analysis to assess the association between wealth index, and previous circumcision towards women's attitude to discontinue the practice of FGM by controlling confounders. To declare a statistically significant association, adjusted odds ratios (AOR) with 95% confidence interval (CI) were used.

## Ethics approval and consent to participate

The study does not required ethical approval because it was a secondary data analysis using the 2016 EDHS database. After receiving the data from the USAID–DHS program, the researchers in this study maintained the data's anonymity. During the survey, consent was received from the study participants prior to the start of study.

## Results

### Description of the study variables with attitude to discontinuation of FGM

A total of 6984 reproductive-aged women were included and analysed. In terms of age, 35.4% of women aged 20–29 years had an attitude toward discontinuation of female genital mutilation. When it came to circumcision, 65.8% of those who supported ending FGM had been circumcised at some point in their lives (Table 1).

### Association between wealth index and women's attitude to discontinue the practice of FGM in Ethiopia

Because all of the variables in bivariate analysis had a p-value of less than 0.25, they were all incorporated into multivariable logistic regression analysis. The odds of attitude towards the discontinuation of female genital mutilation was 1.46 (AOR: 1.46; 95% CI: 1.02–2.01), 1.95 (AOR:1.95; 95% CI: 1.36–2.79), 2.22 (AOR: 2.22; 95% CI: 6.68–31.4), and 2.65 (AOR: 2.65; 95% CI: 1.57–4.46) times higher among women who had a poorer, middle, richer, and richest wealth index quintile, respectively compared to women who were in the poorest quintile.

Besides, the odds of attitude towards the discontinuation of female genital mutilation was 1.57 (AOR: 1.57; 95% CI: 1.17–2.12), 3.38 (AOR: 3.38; 95% CI: 1.96–5.83), 14.5(AOR: 14.5; 95% CI: 6.68–31.4) times higher among women attained primary, secondary, and higher education, respectively compared to non-educated one.

In addition, circumcised women compared to non-circumcised women lower the odds of the attitude towards the discontinuation of female genital mutilation by 78% (AOR: 0.22; 95% CI: 0.15–0.32). Besides, the odds of the attitude towards the discontinuation of female genital mutilation was lower by 66% (AOR: 0.34; 95% CI: 0.22–0.50) among women living in the Afar region, by 38% (AOR: 0.42; 95% CI: 0.27–0.65) among women living in the Somali region, by

**Table 1. Univariate and bivariate analysis of the study variables with attitude to female genital mutilation among reproductive-age women in Ethiopia (n = 6984).**

| Variables | Category | Overall | Attitude to FGM | | p-value |
|---|---|---|---|---|---|
| | | | Discontinued | Continued | |
| | | n (wt.%) | n (wt.%) | n (wt.%) | |
| Age | 15–19 | 1523(21.4) | 1225(22.5) | 298(16.4) | 0.004 |
| | 20–29 | 2547(35.2) | 1992(35.4) | 555(34.4) | |
| | 30–39 | 1879(28.3) | 1461(27.3) | 418(32.7) | |
| | 40–49 | 1035(15.1) | 789(14.7) | 246(16.6) | |
| Residence | Urban | 2543(23.4) | 2284(26.5) | 259(9.6) | <0.001 |
| | Rural | 4441(76.6) | 3183(73.5) | 1258(90.4) | |
| Religion | Non-Muslim | 5862(76.5) | 4462(75.2) | 1400(82.7) | 0.023 |
| | Muslim | 1122(23.5) | 1005(24.8) | 117(17.3) | |
| Marital Status | Never Married | 1942(25.8) | 1674(28.0) | 268(15.4) | <0.001 |
| | Married | 4333(64.7) | 3209(62.3) | 1124(75.8) | |
| | Separated/divorced | 709(9.5) | 584(9.7) | 125(8.8) | |
| Education | No Education | 3011(45.9) | 1959(41.2) | 1052(67.0) | <0.001 |
| | Primary | 2325(35.1) | 1953(36.5) | 372(28.9) | |
| | Secondary | 1058(12.6) | 985(14.6) | 73(3.7) | |
| | Higher | 590(6.4) | 570(7.7) | 20(0.4) | |
| Wealth index | Poorest | 1605(15.4) | 853(12.2) | 752(29.9) | <0.001 |
| | Poorer | 862(17.5) | 659(16.5) | 203(22.0) | |
| | Middle | 900(19.5) | 730(19.4) | 170(19.8) | |
| | Richer | 910(19.9) | 772(20.6) | 138(16.4) | |
| | Richest | 2707(27.8) | 2453(31.3) | 254(11.8) | |
| Access to media | No | 3426(54.1) | 2353(51.1) | 1073(68.0) | <0.001 |
| | Yes | 3558(45.9) | 3114(48.9) | 444(32.0) | |
| Ever been Circumcised | No | 2113(29.4) | 2015(34.2) | 98(7.7) | <0.001 |
| | Yes | 4871(70.6) | 3452(65.8) | 1419(92.3) | |
| Region | Tigray | 658(6.4) | 604(7.1) | 54(2.9) | <0.001 |
| | Afar | 579(0.9) | 226(0.5) | 353(2.9) | |
| | Amhara | 739(22.8) | 601(22.9) | 138(22.4) | |
| | Oromiya | 876(38.0) | 712(37.1) | 164(41.7) | |
| | Somali | 692(3.2) | 312(1.9) | 380(9.4) | |
| | Benishangul- Gumuz | 475(1.0) | 431(1.1) | 44(0.5) | |
| | SNNPR | 786(20.1) | 670(20.6) | 116(17.6) | |
| | Gambela | 322(0.2) | 296(0.2) | 26(0.1) | |
| | Harari | 414(0.3) | 350(0.3) | 64(0.2) | |
| | Addis Ababa | 894(6.5) | 861(7.7) | 33(1.4) | |
| | Dire Dawa | 549(0.6) | 404(0.6) | 145(0.9) | |

59% (AOR: 0.51; 95% CI: 0.32–0.83) among women live in Dire Dawa region compared to Oromiya region. However, the odds of the attitude towards the discontinuation of female genital mutilation was higher among women living in Benishangul- Gumuz by 2.22 times (AOR: 2.22; 95% CI: 1.46–3.36), and Addis Ababa by 2.13 times (AOR: 2.13; 95% CI: 1.13–4.04) (Table 2).

## Discussion

In Ethiopia, the current study looked into whether women with higher wealth indexes have a higher possibility of opposing FGM. Studies from Guinea [16] and other countries [15]

**Table 2. Multivariable analysis using binary logistic regression to identify independent predictors of attitude toward discontinuation of female genital mutilation among reproductive-age women in Ethiopia (n = 6984).**

| variables | category | Attitude to discontinuation FGM | |
|---|---|---|---|
| | | COR(95% CI) | AOR(95%CI) |
| Age | 15–19 | 1 | 1 |
| | 20–29 | 0.75(0.57–0.97) | 1.08(0.77–1.52) |
| | 30–39 | 0.61(0.45–0.81) | 1.25(0.82–1.94) |
| | 40–49 | 0.65(0.48–0.86) | 1.32(0.89–1.94) |
| Residence | Urban | 3.39(2.49–4.60) | 1.01(0.62–1.63) |
| | Rural | 1 | 1 |
| Religion | Non-Muslim | 1 | 1 |
| | Muslim | 1.58(1.06–2.36) | 1.45(0.98–2.13) |
| Marital Status | Never Married | 1 | 1 |
| | Married | 0.45(0.35–0.58) | 0.96(0.67–1.36) |
| | Separated/Divorced | 0.61(0.42–0.87) | 1.08(0.69–1.69) |
| Education | No education | 1 | 1 |
| | Primary | 2.05(1.63–2.58) | 1.57(1.17–2.12)* |
| | Secondary | 6.42(4.03–10.2) | 3.38(1.96–5.83)* |
| | Higher | 30.5(16.3–57.5) | 14.5(6.68–31.4)* |
| Wealth Index | Poorest | 1 | 1 |
| | Poorer | 1.83(1.33–2.53) | 1.46(1.02–2.01)* |
| | Middle | 2.40(1.75–3.28) | 1.95(1.36–2.79)* |
| | Richer | 3.08(2.27–4.17) | 2.22(1.55–3.19)* |
| | Richest | 6.48(4.65–9.03) | 2.65(1.57–4.46)* |
| Access to media | No | 1 | 1 |
| | Yes | 2.04(1.64–2.53) | 0.85(0.65–1.13) |
| Ever been Circumcised | No | 1 | 1 |
| | Yes | 0.16(0.11–0.23) | 0.22(0.15–0.32)* |
| Region | Tigray | 2.75(1.65–4.55) | 1.47(0.84–2.58) |
| | Afar | 0.19(0.14–0.28) | 0.34(0.22–0.50)* |
| | Amhara | 1.14(0.78–1.66) | 0.96(0.68–1.37) |
| | Oromiya | 1 | 1 |
| | Somali | 0.22(0.15–0.33) | 0.42(0.27–0.65)* |
| | Benishangul- Gumuz | 2.27(1.43–3.61) | 2.22(1.46–3.36)* |
| | SNNPR | 1.31(0.83–2.06) | 0.95(0.63–1.44) |
| | Gambela | 2.91(1.82–4.64) | 1.22(0.72–2.07) |
| | Harari | 1.26(0.80–1.99) | 1.02(0.65–1.57) |
| | Addis Ababa | 6.30(3.93–10.1) | 2.13(1.13–4.04)* |
| | Dire Dawa | 0.74(0.49–1.10) | 0.51(0.32–0.83)* |

*-showed significant association with AOR with a 95% confidence interval at a p-value of < 0.05

confirm this conclusion. This may be due to the increment of decision-making authority by monetarily empowered women [19].

This study also adds to the literature that educated women, and regional variation were significant predictors of women's attitude to discontinue the practice of FGM in Ethiopia. Educational attainment of the women results in a higher odds of the attitude to discontinue female genital cutting. This finding is consistent with studies conducted in different places [5, 12–14]. This may be explained as educated women may be equipped with the knowledge to evaluate

their beliefs about traditional practices, and it provides women with financial independence and empowerment to liberate themselves from harmful practices. Furthermore, the opinions of educated women are less likely to be influenced and shaped by traditions.

In comparison to uncircumcised women, circumcised women were less likely to support ending FGM. This result is consistent with the finding of a prior investigation in Ethiopia [13], This can be explained by the fact that having instilled in one's upbringing might give FGM a deeper meaning and purpose. Circumcised women may have gained perks such as recognition and respect as a result of the procedure. However, this conclusion contradicts a study from Egypt [11], which found that previous circumcision leads to cessation of FGM. The discrepancy between the Egyptian study and the current study is attributable to changes in the study participants. The Egyptian study involved medical students whereas the current study involved reproductive-age women residing in the community. Because they were medical students, they were able to predict the health consequences of FGM more easily, which boosted their enthusiasm for the practice's abolition.

Furthermore, women in Afar, Somalia, and Dire Dawa had a reduced likelihood of having a favourable attitude regarding the abolition of female genital mutilation. Women in Benishangul-Gumuz and Addis Ababa, on the other hand, had a more favourable attitude toward the cessation of female genital mutilation. This finding is consistent with previous studies conducted in Ethiopia [20, 21]. This could be due to significant spatio-temporal variation of FGM practice across the country [20].

## Strength and limitation of the study

Although findings of this study are valuable for policymaking, there are a few limitations to be aware of. The data, for example, were gathered from secondary sources, and our study may not be free of the flaws that come with this method. Despite these limitations, our study is one of the few that have contributed to the discussion of the association between wealth index and women's attitude to discontinue the practice of FGM in Ethiopia, as no other studies have been conducted in the country using the most recent 2016 national representative data set available to date.

## Conclusions

The present study revealed that women's wealth index, educated women, previous circumcision, and regional variation significantly predict women's attitude to discontinue the practice of FGM in Ethiopia. Therefore, efforts need to be done in promoting women's socioeconomic status concurrently empowering women in education. Furthermore, designing interventions that address FGM practices should focus on circumcised women along with addressing regional variation to have a significant effect on curtailing this harmful traditional practice from Ethiopia.

## Acknowledgments

We would like to express gratitude to the USAID–DHS program for offering the 2016 Ethiopia Demographic and Health Survey data set.

## Author Contributions

**Conceptualization:** Mohammed Ahmed, Abdu Seid, Seada Seid.

**Data curation:** Mohammed Ahmed.

**Formal analysis:** Mohammed Ahmed.

**Methodology:** Mohammed Ahmed.

**Software:** Mohammed Ahmed, Abdu Seid.

**Validation:** Mohammed Ahmed.

**Visualization:** Mohammed Ahmed, Abdu Seid.

**Writing – original draft:** Mohammed Ahmed, Abdu Seid, Seada Seid.

**Writing – review & editing:** Mohammed Ahmed, Abdu Seid, Seada Seid, Ali Yimer.

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
