## [Decision Letter · Decision Letter 0]

18 Apr 2022

PONE-D-21-07908

Does previous circumcision and wealth index influencing women’s attitude to discontinue the practice of female genital mutilation and cutting (FGM/C) in Ethiopia?

PLOS ONE

Dear Dr. Ahmed,

Thank you for submitting your manuscript to PLOS ONE. After careful consideration, we feel that it has merit but does not fully meet PLOS ONE’s publication criteria as it currently stands. Therefore, we invite you to submit a revised version of the manuscript that addresses the points raised during the review process.

The manuscript has been evaluated by three reviewers, and their comments are available below.

The reviewers have raised a number of concerns that need attention. They request additional information on methodological aspects of the study, comments on the discussion/introduction and other queries/revisions regarding this manuscript.

Could you please revise the manuscript to carefully address the concerns raised?

We look forward to receiving your revised manuscript.

Kind regards,

Sebastian Shepherd

Associate Editor

PLOS ONE

Journal Requirements:

2. During your revisions, please note that a simple title correction is required: The grammar choice of 'influencing' in the title is not correct, one suggested edit to the title could be - 'Does previous circumcision and wealth index influence women’s attitude to discontinue the practice of female genital mutilation and cutting (FGM/C) in Ethiopia?'. Please ensure this is updated in the manuscript file and the online submission information.

4. We noticed you have some minor occurrence of overlapping text with the following previous publication, which needs to be addressed:

- https://obgyn.pericles-prod.literatumonline.com/doi/10.1016/j.ijgo.2009.07.022

The text that needs to be addressed involves the Discussion section.

In your revision ensure you cite all your sources (including your own works), and quote or rephrase any duplicated text outside the methods section. Further consideration is dependent on these concerns being addressed.

Additional Editor Comments (if provided):

Not up to the mark for sending out for review.

Reviewers' comments:

Reviewer's Responses to Questions

**Comments to the Author**

1. Is the manuscript technically sound, and do the data support the conclusions?

Reviewer #1: Yes

Reviewer #2: Yes

Reviewer #3: No

2. Has the statistical analysis been performed appropriately and rigorously? 

Reviewer #1: Yes

Reviewer #2: Yes

Reviewer #3: No

3. Have the authors made all data underlying the findings in their manuscript fully available?

Reviewer #1: Yes

Reviewer #2: Yes

Reviewer #3: Yes

4. Is the manuscript presented in an intelligible fashion and written in standard English?

Reviewer #1: No

Reviewer #2: Yes

Reviewer #3: Yes

5. Review Comments to the Author

Reviewer #1: Thank you so much for submitting this manuscript to PLOS ONE. The manuscript focused on a very important women’s health issue of global relevance for which health providers are in the forefront for care provision and advocacy.

It was indeed a great delight to reviewing this manuscript which has been well conceived and written. Well-done to the team. To further improve on the quality of the manuscript and make it publishable for international audience, below are my suggestions:

General comment on the manuscript.

FGM/C and FGM acronyms and circumcision were inconsistently used in the manuscript. Please make up your mind on the one you want to use and be consistent with its use throughout the manuscript. You also need to clarify that female genital mutilation/cutting is also known as female circumcision so that this becomes clearer for lay global audience.

The manuscript will benefit from editing from primary speakers of English language.

Title: This looks clumsy and unclear. I suggest that you write it out in a clear statement. If you insist on presenting it as a question, kindly revise to ensure clarity.

Abstract

Conclusion: The 1st sentence is not complete. Kindly revise

Background

As previously indicated, you need clearly clarify female genital mutilation/cutting is also known as female circumcision so that this becomes clearer for lay global audience. For example, in paragraph 4, 1st sentence and the entire paragraph you used circumcision for the first time despite previous use of female genital mutilation/cutting. Be consistent throughout the manuscript.

Methodology

Data source: 3rd sentence “A detailed description… were founded elsewhere (2)” Do you mean were reported somewhere?

How did you arrive at your sample size? Any statistical power analysis?

Study variables: You stated that the wealth index were five, however, you mentioned 4- poorest, poorer, middle richer, and richest. Please check and amend.

Ethics Analysis

How did you contact and obtain consent from 6984 women of reproductive age whose data you used? This needs to be clearly presented.

Results

Well presented, thank you.

Discussion

You stated in your findings that “Also, circumcised women compared to non-circumcised women lower the odds of the attitude towards the discontinuation of female genital mutilation by 78%” suggesting that women with FGM/C are more likely to want to continue the practice. However, in the 2nd paragraph of your discussion, you included previous FGM/C as a predictor for discontinuing the practice. Please check and amend.

Your conflicting discussion of your finding is further depicted in the 3rd paragraph where the content here is different to that of paragraph 2 with reference previous FGM/C and attitude to continue or discontinue FGM/C. Please check and amend.

In general, thank you for this manuscript. Best wishes.

Reviewer #2: Corrections about References (See revised Manuscript)

Reviewer #3: The topic of what explains the persistence of FGC, and attitudes in support of FGC, is an important one. The authors use one survey in one country to examine the correlation between FGC attitudes and wealth and education. Unfortunately I do not see how this piece adds to our knowledge of the perpetuation of FGC above and beyond works that have already been published. The authors note that there are some mixed results regarding socioeconomic status and the practice of FGC. However this study does not help us resolve or make sense of these conflicting results but rather adds another case, which has also been included in past studies. It does not attempt to help us theoretically understand the reason for these conflicting results or present results that help in this regard either. As such, I cannot recommend the article for publication as I do not see it contributing new knowledge to the study of FGC, however I commend the authors for their effort. Perhaps including more countries and over a longer period of time, both of which are possible with the DHS data, could help. Note however that a number of scholars have also already done this.

6. PLOS authors have the option to publish the peer review history of their article (what does this mean?). If published, this will include your full peer review and any attached files.

Reviewer #1: **Yes: **Dr Olayide Ogunsiji

Reviewer #2: No

Reviewer #3: No

---

## [Author Response · Author response to Decision Letter 0]

26 Apr 2022

Thank you very much for PLOS one editorial office, academic editors, as well as reviewers of this manuscript entitled Does previous circumcision and wealth index influence women’s attitude to discontinue the practice of female genital mutilation in Ethiopia? For their astonished effort.

The written documents below explained point by point response for respective editor and reviewer’s comment.

Editor comments and author response 

During your revisions, please note that a simple title correction is required: The grammar choice of 'influencing' in the title is not correct, one suggested edit to the title could be - 'Does previous circumcision and wealth index influence women’s attitude to discontinue the practice of female genital mutilation and cutting (FGM/C) in Ethiopia?'. 

Author response: Thank you so much. The authors amended the manuscript based on the reviewer comments and included in the revised manuscript.

Reviewer 1 comments and authors response: 

Reviewer #1: 

General comment on the manuscript.

FGM/C and FGM acronyms and circumcision were inconsistently used in the manuscript. Please make up your mind on the one you want to use and be consistent with its use throughout the manuscript. You also need to clarify that female genital mutilation/cutting is also known as female circumcision so that this becomes clearer for lay global audience. The manuscript will benefit from editing from primary speakers of English language.

Author response: Thank you so much. The authors amended the manuscript based on the reviewer comments and included in the revised manuscript.

Title: This looks clumsy and unclear. I suggest that you write it out in a clear statement. If you insist on presenting it as a question, kindly revise to ensure clarity.

Author response: The authors amended the manuscript based on the reviewer comments and included in the revised manuscript.

Abstract

Conclusion: The 1st sentence is not complete. Kindly revise

Author response: The authors amended the manuscript based on the reviewer comments and included in the revised manuscript.

Background

As previously indicated, you need clearly clarify female genital mutilation/cutting is also known as female circumcision so that this becomes clearer for lay global audience. For example, in paragraph 4, 1st sentence and the entire paragraph you used circumcision for the first time despite previous use of female genital mutilation/cutting. Be consistent throughout the manuscript.

Author response: The authors amended the manuscript based on the reviewer comments and included in the revised manuscript.

Methodology

Data source: 3rd sentence “A detailed description… were founded elsewhere (2)” Do you mean were reported somewhere? How did you arrive at your sample size? Any statistical power analysis?

Study variables: You stated that the wealth index were five, however, you mentioned 4- poorest, poorer, middle, richer, and richest. Please check and amend.

 Author response: The authors amended the manuscript based on the reviewer comments and included in the revised manuscript. In the 2016 EDHS, 15,683 women (15–49 years) were interviewed, For studying FGM, after excluding cases with missing values on the variables of interests, the analytic sample in this study consisted of 6984 reproductive age women (15- 49 years).

Ethics Analysis

How did you contact and obtain consent from 6984 women of reproductive age whose data you used? This needs to be clearly presented.

Author response: The authors amended the manuscript based on the reviewer comments and included in the revised manuscript.

Results

Well presented, thank you.

Discussion

You stated in your findings that “Also, circumcised women compared to non-circumcised women lower the odds of the attitude towards the discontinuation of female genital mutilation by 78%” suggesting that women with FGM/C are more likely to want to continue the practice. However, in the 2nd paragraph of your discussion, you included previous FGM/C as a predictor for discontinuing the practice. Please check and amend.

Author response: The authors amended the manuscript based on the reviewer comments and included in the revised manuscript.

Your conflicting discussion of your finding is further depicted in the 3rd paragraph where the content here is different to that of paragraph 2 with reference previous FGM/C and attitude to continue or discontinue FGM/C. Please check and amend.

In general, thank you for this manuscript. Best wishes.

Author response: The authors amended the manuscript based on the reviewer comments and included in the revised manuscript. 

Reviewer 2 comments and author response 

Reviewer #2: Corrections about References (See revised Manuscript)

Author response: The authors amended the manuscript based on the reviewer comments and included in the revised manuscript. 

Reviewer #3: The topic of what explains the persistence of FGC, and attitudes in support of FGC, is an important one. The authors use one survey in one country to examine the correlation between FGC attitudes and wealth and education. Unfortunately I do not see how this piece adds to our knowledge of the perpetuation of FGC above and beyond works that have already been published. The authors note that there are some mixed results regarding socioeconomic status and the practice of FGC. However this study does not help us resolve or make sense of these conflicting results but rather adds another case, which has also been included in past studies. It does not attempt to help us theoretically understand the reason for these conflicting results or present results that help in this regard either. As such, I cannot recommend the article for publication as I do not see it contributing new knowledge to the study of FGC, however I commend the authors for their effort. Perhaps including more countries and over a longer period of time, both of which are possible with the DHS data, could help. Note however that a number of scholars have also already done this

Author response: There was no study conducted so far in relation to wealth index and attitude to discontinue FGM in Ethiopia. That is why we eager to undertake this study. 

Thank you

---

## [Decision Letter · Decision Letter 1]

8 Jul 2022

PONE-D-21-07908R1Does previous circumcision and wealth index influence women’s attitude to discontinue the practice of female genital mutilation and cutting (FGM/C) in Ethiopia?PLOS ONE

Dear Dr. Ahmed,

Thank you for submitting your manuscript to PLOS ONE. After careful consideration, we feel that it has merit but does not fully meet PLOS ONE’s publication criteria as it currently stands. Therefore, we invite you to submit a revised version of the manuscript that addresses the points raised during the review process.

The manuscript has been evaluated by one reviewer and their comments are available below.

The reviewer has raised a concern that the manuscript still requires extensive copyediting. We suggest you thoroughly copyedit your manuscript for language usage, spelling, and grammar. If you do not know anyone who can help you do this, you may wish to consider employing a professional scientific editing service.  

Could you please revise the manuscript to carefully address the concerns raised?

We look forward to receiving your revised manuscript.

Kind regards,

Sebastian Shepherd

Staff Editor

PLOS ONE

Journal Requirements:

Reviewers' comments:

Reviewer's Responses to Questions

**Comments to the Author**

1. If the authors have adequately addressed your comments raised in a previous round of review and you feel that this manuscript is now acceptable for publication, you may indicate that here to bypass the “Comments to the Author” section, enter your conflict of interest statement in the “Confidential to Editor” section, and submit your "Accept" recommendation.

Reviewer #1: All comments have been addressed

2. Is the manuscript technically sound, and do the data support the conclusions?

Reviewer #1: Yes

3. Has the statistical analysis been performed appropriately and rigorously? 

Reviewer #1: Yes

4. Have the authors made all data underlying the findings in their manuscript fully available?

Reviewer #1: Yes

5. Is the manuscript presented in an intelligible fashion and written in standard English?

Reviewer #1: Yes

6. Review Comments to the Author

Reviewer #1: Thank you for responding to majority of my comments in the previous version of this manuscript. However, the manuscript will still benefit from editing done by a primary speaker of English language. This is important for the international audience that this journal targets.

7. PLOS authors have the option to publish the peer review history of their article (what does this mean?). If published, this will include your full peer review and any attached files.

Reviewer #1: No

---

## [Author Response · Author response to Decision Letter 1]

11 Jul 2022

Thank you very much for PLOS one editorial office, academic editors, as well as reviewers of this manuscript entitled Does previous circumcision and wealth index influence women’s attitude to discontinue the practice of female genital mutilation and cutting (FGM/C) in Ethiopia?.

The written documents below explained point by point response for respective reviewer and editor comments.

Reviewer 1 comments and authors response: 

Comment 1: the reviewer raised a concern that the manuscript still requires extensive copyediting. We suggest you thoroughly copyedit your manuscript for language usage, spelling, and grammar.

Author response: the authors addressed all of the issue raised by the reviewer in the revised manuscript. The copy-editing was done by the authors.

Editor comments and author response

Author response: I have reviewed the references in the manuscript and I did not found retracted articles. 

Thank you for reviewers and editors (staff as well as academic editors) for their extensive and constructive comments throughout the process

---

## [Editor Report · Decision Letter 2]

1 Aug 2022

Does previous circumcision and wealth index influence women’s attitude to discontinue the practice of female genital mutilation and cutting (FGM/C) in Ethiopia?

PONE-D-21-07908R2

Dear Dr. Ahmed,

We’re pleased to inform you that your manuscript has been judged scientifically suitable for publication and will be formally accepted for publication once it meets all outstanding technical requirements.

Kind regards,

Sebastian Shepherd

Staff Editor

PLOS ONE

**Additional Editor Comments (optional):**

Please note, we have edited your abstract for clarity and style (the edited version is below - and we have also attached a copy of the edited abstract as a word file). Please review these changes and incorporate those that you agree with for the final version of the manuscript.

***** Abstract Edit *****

Introduction: Understanding women’s attitudes towards female genital mutilation is an important step towards eliminating this practice. We used the 2016 Ethiopia Demographic and Health Survey (EDHS) data set to examine the relationship between wealth index and previous history of circumcision on women's opinions whether female genital mutilation (FGM) should be continued or stopped in Ethiopia.

Methods: Data from 6984 women aged 15-49 years were extracted from the 2016 EDHS data set. Multivariable logistic regression analysis was performed to analyse the data.

Result: In this study, women with a higher level of education and wealth index were more likely to support the cessation of FGM. However, circumcised women (AOR: 0.22; 95% CI: 0.15-0.32), women from the Afar region (AOR: 0.34; 95% CI: 0.22-0.50), Somali region (AOR: 0.42; 95% CI: 0.27-0.65), and Dire Dawa region (AOR: 0.51; 95% CI: 0.32-0.83) were less likely to support discontinuation of FGM.

Conclusion: The present study revealed that wealth index, education level, history of circumcision, and regional variation are associated with women’s attitude towards discontinuation of the practice of FGM in Ethiopia. Empowering women in terms of socioeconomic status and education can change attitudes and might help prevent female genital mutilation in the future. Furthermore, interventions targeting FGM practices should focus on regional variance in order to have a meaningful impact on reducing this harmful cultural practice in Ethiopia.
---

## [Editor Report · Acceptance letter]

11 Aug 2022

PONE-D-21-07908R2 

Does previous circumcision and wealth index influence women’s attitude to discontinue the practice of female genital mutilation and cutting (FGM/C) in Ethiopia? 

Dear Dr. Ahmed:

I'm pleased to inform you that your manuscript has been deemed suitable for publication in PLOS ONE. Congratulations! Your manuscript is now with our production department. 

Kind regards, 

on behalf of

Dr Sebastian Shepherd 

Staff Editor

PLOS ONE